# Target-Site Mutations and Glutathione *S*-Transferases Are Associated with Acequinocyl and Pyridaben Resistance in the Two-Spotted Spider Mite *Tetranychus urticae* (Acari: Tetranychidae)

**DOI:** 10.3390/insects11080511

**Published:** 2020-08-07

**Authors:** Jihye Choi, Hyun-Na Koo, Sung Il Kim, Bueyong Park, Hyunkyung Kim, Gil-Hah Kim

**Affiliations:** 1Department of Plant Medicine, College of Agriculture, Life and Environment Sciences, Chungbuk National University, Cheongju 28644, Korea; cjh5767@gmail.com (J.C.); hyunnakoo@hanmail.net (H.-N.K.); wmwl010@naver.com (S.I.K.); nshk0917@hanmail.net (H.K.); 2Crop Protection Division, National Institute of Agricultural Science, Wanju 55365, Korea; florigen1@korea.kr

**Keywords:** *Tetranychus urticae*, acequinocyl, pyridaben, point mutation, glutathione *S*-transferase

## Abstract

**Simple Summary:**

The two-spotted spider mite *Tetranychus urticae* is a difficult-to-control pest due to its short life cycle and rapid resistance development. In this study, we characterized field strains collected in 2001 and 2003 that have been selected for acequinocyl resistance and pyridaben resistance, respectively. These strains displayed resistance ratios of 1798.6 and 5555.6, respectively, and were screened for cross-resistance against several currently used acaricides. The acequinocyl resistant strain exhibited pyridaben cross-resistance, but the pyridaben resistant strain showed no cross-resistance. The acequinocyl resistant strain exhibited point mutations in *cytb* (I256V and N321S) and PSST (H92R). In contrast, the pyridaben resistant strain exhibited the H92R but not the I256V and N321S point mutations. In addition, the increased GST metabolism and GST delta expression might be related to acequinocyl resistance in *Tetranychus urticae*. We hope that the data and patterns described here can now be exploited in the continued quest for rational resistance management strategies.

**Abstract:**

The two-spotted spider mite *Tetranychus urticae* is a difficult-to-control pest due to its short life cycle and rapid resistance development. In this study, we characterized field strains collected in 2001 and 2003 that were selected for acequinocyl resistance (AR) and pyridaben resistance (PR), respectively. These strains displayed resistance ratios of 1798.6 (susceptible vs. AR) and 5555.6 (susceptible vs. PR), respectively, and were screened for cross-resistance against several currently used acaricides. The AR strain exhibited pyridaben cross-resistance, but the PR strain showed no cross-resistance. The AR strain exhibited point mutations in *cytb* (I256V, N321S) and PSST (H92R). In contrast, the PR strain exhibited the H92R but not the I256V and N321S point mutations. In some cases increased glutathione *S*-transferase (GST) activity has previously been linked to enhanced detoxification. The AR strain exhibited approximately 2.3-, 1.8-, and 2.2-fold increased GST activity against 1-chloro-2,4-dinitrobenzene (CDNB), 1,2-dichloro-4-nitrobenzene (DCNB), and 4-nitrobenzyl chloride (NBC), respectively. Among the five GST subclass genes (delta, omega, mu, zeta, and kappa), the relative expression of delta class GSTs in the AR strain were significantly higher than the PR and susceptible strain. These results suggest that the I256V and N321S mutations and the increased GST metabolism and GST delta overexpression might be related to acequinocyl resistance in *T. urticae*.

## 1. Introduction

The two-spotted spider mite *Tetranychus urticae* Koch (Arthropoda: Acari: Tetranychidae) is a worldwide agricultural pest that invades a wide range of host crops. The management of *T. urticae* is mainly achieved through chemical control [1,2,3,4]. *Tetranychus urticae* is likely to develop acaricide resistance faster than other pests, and there have indeed been reports on its development of resistance to almost all types of acaricides currently registered [5,6]. Acequinocyl is a naphthoquinone compound discovered in the 1970s by DuPont [7]. It is a proacaricide that breaks down into the active metabolite, a deacetylated product. A mechanistic study showed that the deacetylated metabolite of acequinocyl inhibits respiration in mitochondria at the ubiquinol oxidation site (Q_0_) of complex III of the electron transfer chain [8,9,10]. Pyridaben is a novel pyridazine compound discovered in 1984 by Nissan Chemical and commercialized in 1991 [11,12,13]. The compound affects metabolism, inhibiting the mitochondrial electron transport chain by binding complex I at the coenzyme site Q_0_ [14]. Pyridaben specifically blocks mitochondrial and isolated complex I oxidations with high potency. 

Arthropods have developed two main mechanisms of acaricide resistance: decreasing exposure due to quantitative or qualitative changes in major detoxification enzymes and decreasing sensitivity due to changes in target-site sensitivity caused by point mutations [15,16,17]. Fotoukkiaii et al. [18] demonstrated that amino acid residue substitutions (G126S + A133T) in cytochrome *b* of *T. urticae* are associated with acequinocyl resistance. Kim et al. [19] also observed the I256V + N321S point mutations in an acequinocyl-resistant strain. However, the G126S + A133T point mutations were detected in field-collected populations of *T. urticae*, but the I256V + N321S point mutations were not found. These findings suggest that alternative mechanisms are in place, possibly including increased metabolism, as was previously reported. A H92R substitution was identified in the NADH ubiquinone oxidoreductase subunit PSST (PSST) homologue of METI (mitochondrial electron transport inhibitors)-I acaricide resistant strains. PSST is a subunit of NADH dehydrogenase (ubiquinone), also known as Complex I, which is located in the mitochondrial inner membrane and is the largest of the five complexes of the electron transport chain.

Many studies in the past have linked an increase in detoxifying enzyme activities to a certain resistant phenotype. Genetically fixed resistance mechanisms in spider mites are thought to be similar to those documented in insects and involve enhanced detoxification through the enzymatic activity of esterases (ESTs), glutathione-S-transferases (GSTs) and cytochrome P450 monooxygenases (P450s) [20]. Nieuwenhuyse et al. (2009) found that the P450 enzyme was higher compared to the susceptible population in an acequinocyl-resistant population of *T. urticae* [21]. Salman et al. (2014) found that the GST and P450 enzymes do not appear to have any significant involvement [10]. Furthermore, rapid monitoring can effectively control resistant pests. The spray method and the dipping method are general bioassay methods commonly used for insecticide-resistance monitoring but have the disadvantage of being time-consuming. Recently, to address this disadvantage, resistance monitoring related to mechanisms of action using quantitative trait loci (QTL), PCR amplification of specific alleles (PASA), quantitative sequencing (QS), and serial invasive signal amplification reactions has been conducted [22,23].

In this study, we evaluated the susceptibility of laboratory-selected (acequinocyl- and pyridaben-resistant) strains of *T. urticae* to twelve commercial acaricides and analyzed their target-site mutations (G126S + A133T, I256V + N321S and H92R). In addition, we identified the activities of detoxification enzymes and determined the expression levels of five GST subclass genes (delta, omega, mu, zeta, and kappa).

## 2. Materials and Methods

### 2.1. T. urticae Strains

The susceptible (S) strain of *T. urticae* used in this study was reared, beginning in 2005, in a laboratory at Chungbuk National University (Cheongju, Korea). The two resistant *T. urticae* populations were collected from glasshouse-cultivated roses in Gimhae (Gyungnam, Republic of Korea, in 2001) and Uiseong (Gyungbuk, Korea, in 2003). These populations were treated once a week with acequinocyl and pyridaben, respectively, at a range of concentrations corresponding to their LC_30_-LC_50_ values and selected over sixteen years for acequinocyl (named AR) and pyridaben resistance (named PR). The mites were reared at 25–27 °C under 40–60% relative humidity and a 16:8 (L:D) photoperiod. Potted kidney beans (*Phaseolus vulgaris* L.) were used as a host.

### 2.2. Acaricides

Commercially formulated abamectin (1.8% EC), acequinocyl (15% SC), azocyclotin (25% WP), bifenazate (13.5% SC), cyenopyrafen (25% SC), cyflumetofen (20% SC), etoxazole (10% SC), milbemectin (1% EC), pyflubumide (10% SC), pyridaben (20% WP), spirodiclofen (22% WP), and spiromesifen (20% SC) were purchased from a farm supply store (Seowon, Cheongju, Korea).

### 2.3. Toxicological Assay in the Laboratory 

#### 2.3.1. *T. urticae* Females

Briefly, we tested a minimum of five concentrations in four replicates. For each replicate, 20–30 adult females (2 to 3 days old) were transferred to bean leaf disks (35 mm in diameter) on wet cotton wool. Thirty females were transferred to the leaf disk using a brush. Solutions diluted to various concentrations were sprayed (3 mL each) onto the disks, which were then dried in the dark for 30 min. The dish was incubated at 25–27 °C under 40–60% relative humidity and a 16:8 (L:D) photoperiod. Mortality was evaluated at 48 h after treatment. The mortality of the treated insects was corrected using the control mortality, and the corrected data were used to calculate LC_50_ values. Blank controls were sprayed with deionized water only, and control mortality in all tests never exceeded 5%. All experiments were replicated three times.

#### 2.3.2. *T. urticae* Eggs

Bean leaf disks of approximately 35 mm in diameter were used as substrates for oviposition. Four leaf disk were used for each treatment and ten 2- to 3-day-old mated females were placed on the ventral side of the bean leaf disk placed on cotton soaked in water in a Petri dish (60 mm diameter) and 24 h was allowed to lay eggs. After 24 h, the adults were removed and the eggs were counted to get at least 25 eggs per disc. The leaf disk with eggs were treated with the selected pesticide suspension using a sprayer and allowed to shade dry for 30 min. A water spray control was maintained in the experiment. All disks were examined daily for 7 days. The numbers of hatched and non-hatched eggs were recorded. The dish was incubated at 25–27 °C under 40–60% relative humidity and a 16:8 (L:D) photoperiod.

### 2.4. General Sequencing and Pyrosequencing of cytb

Genomic DNA was individually extracted from approximately 100 mites of each *T. urticae* strain using the DNeasy Blood and Tissue kit (Qiagen, Hilden, Germany) according to the manufacturer’s instructions. The template (1 μL) was included in each PCR sample (HotStart PCR Premix kit, Bioneer Co., Daejeon, Korea). The reactions were performed using previously reported primers [21,23]. The resulting PCR products were purified and directly sequenced by Bioneer Co. A recently published pyrosequencing method [24] was optimized for use with genomic DNA. Briefly, a short gene fragment was amplified from 50 ng aliquots of gDNA (adults) via PCR using a new primer pair (Table 1).

The pyrosequencing protocol consisted of 45 PCR cycles performed with the forward primer and biotinylated reverse primer at 0.5 μM each in 20 μL reaction mixtures containing 1× Taq enzyme reaction mix (Enzynomics, Daejeon, Korea). The following cycling conditions were used: one cycle at 95 °C for 10 min; 45 cycles of 95 °C for 30 s, 52/54/56/60 °C for 30 s and 72 °C for 30 s; and a final elongation step at 72 °C for 5 min. The reactions were performed using a PyroGold reagent kit and a PyroMark ID system (Qiagen).

### 2.5. Detoxification Enzyme Assays

The methods of Salman and Sarıtaş [10] were adapted to determine the activities of GSTs, ESTs and P450s. To calculate the GST activity, 1-chloro-2,4-dinitrobenzene (CDNB), 1,2-dichloro-4-nitrobenzene (DCNB), and 4-nitrobenzyl chloride (NBC) were used as the substrates. Fifty mature females were homogenized in 300 μL of Tris–HCl buffer (0.05 M, pH 7.5) in Eppendorf tubes using a plastic pestle, followed by centrifugation at 10,000× *g* and 4 °C for 5 min. The total volume of 300 μL, which consisted of 100 μL of supernatant, 100 μL of CDNB (0.1% v v^−1^ in ethanol), 100 μL of DCNB, and 100 μL of NBC was transferred to the 96-well microplate, respectively. In the microplate cells, 0.4 mM CDNB, DCNB, NBC and GSH were found in the final concentration, respectively. The change in absorbance was calculated at 340 nm and 25 °C for 5 min using the VersaMax kinetic microplate reader (Molecular Devices, Sunnyvale, CA, USA).

For ESTs, α-naphthyl acetate (NA) and β-NA were used as the substrates. The reaction mixture containing 450 μL of 4 mM potassium phosphate buffer (pH 6.8) and 50 μL of enzyme solution (equivalent to 2.5 mites) was incubated at 37 °C for 15 and 5 min after addition of 0.5 mL of 0.5 mM α- and β-NA in ethanol, respectively. The reaction was stopped and color developed by adding 0.5 mL of dye solution (1% diazoblue B salt and 5% sodium lauryl sulfate, 2:5 by volume) for 20 min. The absorbance was read at 600 nm for α-naphthol and at 550 nm for β-naphthol.

To determine the P450 activity, 3,3′,5,5′-tetramethylbenzidine (TMBZ) was used as the substrate. Briefly, homogenate supernatant was diluted to 5 µg protein mL^−1^ in sodium acetate buffer (0.25 M, pH 5.0). The wells of a 96-well microtitre plate were filled with 100 µL of diluted homogenate, 200 µL of 3,3′,5,5′-tetramethylbenzidine (TMBZ) solution and 25 µL of 3% hydrogen peroxide solution. The plate was incubated for 5 min at 25 °C and read at 655 nm in the microplate reader.

All of the assays for the enzyme activities were replicated at least four times. Wells without homogenate served as controls.

### 2.6. Real-Time Quantitative PCR

Total RNA was extracted from approximately 500 female mites per strain, using an easy-spin Total RNA Extraction Kit (iNtRON, Seoul, Korea) according to the manufacturer’s instructions. The extracted RNA was quantified and utilized for qRT-PCR. cDNA was then synthesized using Maxime RT PreMix (iNtRON) following the manufacturer’s instructions. The primers for the five GST subclasses (delta, omega, mu, zeta, and kappa) are listed in Table 1. We selected only one representative gene from each GST subclass. The reactions were performed in 20 μL mixtures containing 10 μL SYBR Premix Ex Taq (2×), 5 pmol/mL forward primer, 5 pmol/mL reverse primer, and 2 μL synthesized cDNA with a Rotor-Gene Q cycler (Qiagen). After incubation at 95 °C for 5 min, 40 PCR cycles (5 s at 95 °C, 10 s at 55 °C, and 15 s at 72 °C) were conducted. Calculations were performed using the Ct values obtained at the end of the PCR via the ΔΔCt method [25]. Based on the equation ΔΔCt = (Ct_target_ − Ct_reference_)_treatment_ − average(Ct_target_ − Ct_reference_)_control_, the expression level of the pretreatment samples (controls; actin) was set to a value of 1, and the results for posttreatment samples represent the fold change relative to expression level of the control sample. In addition, positive or negative ΔΔCt values indicate up- or down-regulation. The real-time PCR analysis was replicated three times in independent biological experiments.

### 2.7. Data Analysis

To estimate the parameters of a concentration-mortality line for each leaf-dip bioassay, replicate data were collected and analyzed using the probit model in the SAS program (SAS Institute 9.3, Cary, NC, USA). Two LC_50_ values were considered different at *p* < 0.01.

## 3. Results

### 3.1. Evaluation of the Resistance Ratio (RR)

The RRs for acequinocyl were 1798.6 and 128.2 for the *T. urticae* AR adults and eggs, respectively. The RRs for pyridaben were 5555.6 and 2739.7 for the *T. urticae* PR adults and eggs, respectively (Table 2 and Table 3). Bioassays were carried out with twelve acaricides on the AR and PR strains of *T. urticae* to evaluate resistance and cross-resistance. Adults of the AR strain showed cross-resistance to pyridaben with an RR of 2777.8, but adults of the PR strain showed relatively low cross-resistance to acequinocyl, with an RR of 4.8. For other acaricides to which *T. urticae* showed cross-resistance, adults of the AR strain showed RRs of 332.6 and 510.2 for etoxazole and spiromesifen, respectively, and eggs of the AR strain showed RRs of 8.0 and 11.5 for cyflumetofen and etoxazole, respectively. Adults of the PR strain showed cross-resistance to cyenopyrafen, cyflumetofen, etoxazole, and spiromesifen, with RRs of 5.8, 5.8, >215.1, and 161.0, respectively, and eggs of the PR strain showed cross-resistance to azocyclotin, cyflumetofen, etoxazole, milbemectin and pyflubumide, with RRs of 11.4, 8.4, >6250, 6.8, and 61.0, respectively. Adults of the AR strain showed negatively correlated cross-resistance to cyenopyrafen and cyflumetofen, with RRs of 0.3 and 0.04, respectively, and the PR strain showed negatively correlated cross-resistance to milbemectin and pyflubumide, with RRs of 0.4 and 0.3, respectively.

### 3.2. Cytochrome b and PSST Genotypes of the Mite Strains

Using pyrosequencing, the frequencies of G126S, A133T, I256V, and N321S in mitochondrial cytochrome *b* and H92R in the PSST subunit of mitochondrial electron transport complex I were identified (Table 4). Two point mutations, I256V and N321S, were found in the AR strain, but G126S and A133T were not detected. The genotype frequencies of a valine (V) at the 256th amino acid position and a serine (S) at the 321st amino acid position were 81.0% and 100.0%, respectively. Interestingly, H92R was detected in the AR strain and had a high genotype frequency of 88%. However, in the PR strain, only the H92R point mutation was identified. The genotype frequency was 94%. The results of allele frequency determination using pyrosequencing were consistent with those of the bioassay.

### 3.3. Detoxifying Enzyme Activities

The in vitro activities of GSTs, nonspecific ESTs, and P450s determined in whole mite homogenates of strains S, AR and PR are presented in Table 5. Only the GST activities measured with CDNB, DCNB, and NBC were significantly different between strains, with a 1.8- to 2.3-fold greater activity in AR than in the sensitive strain.

### 3.4. Expression Levels of Five GST Subclasses

The gene expression levels of five GST subclasses (delta, omega, mu, zeta, and kappa) in the S, AR, and PR strains of *T. urticae* were assessed via qRT-PCR (Figure 1). Among the GST subclasses, the expression levels of only delta GSTs were higher in the AR strain than in the S and PR strains. Therefore, we assumed that the higher expression levels of delta GSTs may be associated with acequinocyl resistance in *T. urticae*.

## 4. Discussion

Two populations of *T. urticae* were selected for 16 years for acequinocyl or pyridaben resistance. They had a high resistance to each of the acaricides. Adults and eggs of the AR strain showed cross-resistance to pyridaben, but adults and eggs of the PR strain showed relatively low cross-resistance to acequinocyl. Other acaricides to which adults of the AR strain showed cross-resistance were etoxazole and spiromesifen. However, eggs of the AR strain were not. Adults of the PR strain showed cross-resistance to cyenopyrafen, cyflumetofen, etoxazole, and spiromesifen, and eggs of the PR strain showed cross-resistance to azocyclotin, cyflumetofen, etoxazole, milbemectin and pyflubumide. Eggs and adults of each strain showed slightly different results to the same acaricide. In addition, the AR and PR strain showed cross-resistance or negatively correlated cross-resistance to the same acaricide. The reason for this is that acaricides have different modes of action and the resistance pattern depends on the pesticide used. Therefore, the effective use of acaricides in the field must be managed. Both acequinocyl and pyridaben are mitochondrial electron transport inhibitors (METIs), but they have different modes of action (targeting mitochondrial complexes III and I, respectively). Therefore, the possibility of cross-resistance to these compounds is low [27,28]. Furthermore, the AR strain was originally derived from fields and may have already developed cross-resistance prior to collection. The S, AR, and PR were from completely distinct genetic backgrounds.

Resistance to acaricide develops due to a number of factors, among which point mutations are the most common. G119S, A201S, T280A, G328A, and F331W/Y point mutations of acetylcholinesterase genes have been reported in the two-spotted spider mite [29]. L1024V, A1215D, and F1538I mutations of voltage-gated chloride channel genes and a I1017F mutation of chitin synthase I have been reported [30,31]. Van Leeuwen et al. [32] reported G126S, I136T, S141T, P262T and A133T, G132A [18] mutations in *cytb* of acequinocyl- and bifenazate-resistant *T. urticae*. In addition, I256V and N321S mutations were reported in *cytb* of acequinocyl-resistant *T. urticae* in a recent study [19]. *Cytb* is an oxidation-reduction protein that acts with *cytc*_1_ and iron-sulfur proteins as a catalyst in mitochondrial complex III [33]. PSST, through which pyridaben exerts activity, is a subunit of mitochondrial complex I. Mitochondrial complex I (NADH: ubiquinone oxidoreductase) is the largest mitochondrial complex and consists of several subunits [34]. The mechanism of catalysis in mitochondrial complex I is not precisely known, but a recent study by Bajda et al. [21] reported that ubiquinone and an inhibitor of complex I bind through PSST and found the H92R point mutation in PSST from pyridaben-resistant *T. urticae*. In this study, the G126S and A133T mutations were not detected in the AR strain, but I256V and N321S were identified. In addition, the H92R mutation was found. Consequently, the H92R mutation is thought to be the reason for cross-resistance to pyridaben in the AR strain of *T. urticae*.

It is reasonable for us to compare the activities of P450s, GSTs, and ESTs between susceptible and resistant *T. urticae* strains, as they are the main enzymes functioning in the detoxification and metabolizing of exogenous chemicals, such as a variety of acaricides [15,35,36]. Our results showed that the activity of P450s and ESTs did not differ significantly between susceptible and resistant *T. urticae*; however, the activities of GSTs were significantly higher (1.81–2.3) in the AR strain of *T. urticae* than in the susceptible strain. Furthermore, increased mRNA levels of GST delta were observed in the AR strain. However, we selected only one gene from each GST class, and further experimentation of the other genes is needed to confirm or reject our findings. 

## 5. Conclusions

Based on the above results, the I256V and N321S mutations and the increased GST metabolism and GST delta expression might be related to acequinocyl resistance in *T. urticae*. Possibly, the over-expressed GSTs are more capable of metabolizing acequinocyl and do not play an important role in the detoxification of pyridaben. We hope that the data and patterns described here can now be exploited in the continued quest for rational resistance management strategies.

## Figures and Tables

**Figure 1 insects-11-00511-f001:**
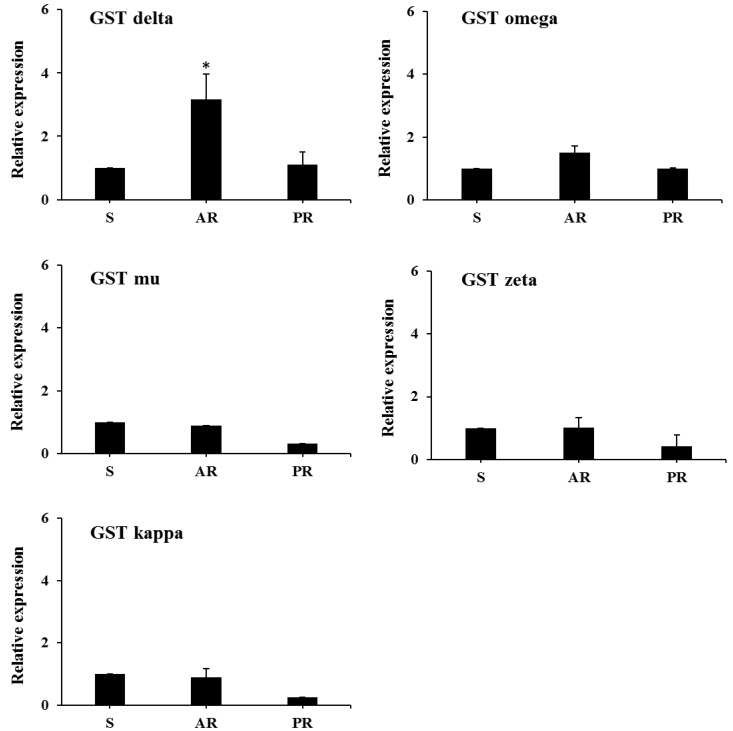
Relative expression levels of GST subclasses in the S, AR, and PR strains of *T. urticae*. The vertical bars indicate standard errors of the means (*n* = 3, 4). * *p* < 0.05 [26].

**Table 1 insects-11-00511-t001:** List of primers used in this study.

Purpose	Gene	Primer Name	Sequence (5′ → 3′)
General PCR	Cytb	PEWY-F	AAAGGCTCATCTAACCAAATAGG
		PEWY-R	AATGAAATTTCTGTAAAAGGGTATTC
	PSST	PSST-F	ACAGGTCAGCCAATCGAATC
		PSST-R	ATACCAAGCCTGAGCAGTGG
Pyrosequencing	Cytb	cytb1-F	TCCAGCTGACCCTCTAAATACAC
		cytb1-R	AGATCGTAGAATTGCGTAAGCAAAT
		cytb2-F	GGAGGAATTTTGAGACTATTAATTCAT
		cytb2-R	ATTTCTGTAAAAGGGTATTCAATT
	PSST	PSST-F	TGACTTTTGGATTAGCCTGTTGTG
		PSST-R	AGGACTTGCTCTGAATAACATACCA
Quantitative RT-PCR	GST	Delta-F	TGGGAAAGTCGGGCAATCAT
		Delta-R	GCACCAAGAGAGGCGTAGAG
		Omega-F	TTGGGAAAGTCGCTCCATCA
		Omega-R	AAACAAGGTTCCTGCATCCCA
		Mu-F	TGGCTCCTGTTCTTGGCTAT
		Mu-R	TCCGGAGCTGGTCCATAGTT
		Zeta-F	ATGGGCGCACCGATTGATT
		Zeta-R	GAACCAAGAACACATCGGCAA
		Kappa-F	AGCTAAAGGGGCTCACTTGAC
		Kappa-R	ACAAAGTCTCCAGCGGCTAT
	Actin	Actin-F	TGTGTGACGACGAAGTAGCC
		Actin-R	AGTCCTTTTGGCCCATACCG

**Table 2 insects-11-00511-t002:** Susceptibility to acaricides in the S, AR and PR strains of *Tetranychus urticae* adults.

Acaricides	Strain	N	LC_50_ (mg/L) (95% CL ^(a)^)	Slope ± SE	RR ^(b)^
Abamectin	S	180	0.14 (0.09–0.21)	1.01 ± 0.10	1.0
	AR	225	0.56 (0.12–0.83)	2.02 ± 0.10	4.0
	PR	225	0.065 (0.02–0.23)	1.71 ± 0.29	0.5
Acequinocyl	S	225	2.78 (1.48–6.58)	0.51 ± 0.07	1.0
	AR	210	>5000	-	>1798.6
	PR	225	13.41 (10.06–21.94)	0.92 ± 0.10	4.8
Azocyclotin	S	225	43.16 (35.63–51.11)	3.06 ± 0.36	1.0
	AR	225	118.17 (85.56–146.64)	1.87 ± 0.28	2.7
	PR	210	20.79 (10.46–33.83)	1.54 ± 0.20	0.5
Bifenazate	S	180	1.12 (0.76–1.66)	2.09 ± 0.47	1.0
	AR	225	4.18 (7.52–18.09)	1.57 ± 0.27	3.7
	PR	210	3.70 (1.80–8.79)	1.08 ± 0.12	3.3
Cyenopyrafen	S	210	0.96 (0.39–3.32)	1.25 ± 0.17	1.0
	AR	180	0.29 (0.18–0.47)	0.75 ± 0.10	0.3
	PR	225	5.57 (2.27–14.76)	0.81 ± 0.10	5.8
Cyflumetofen	S	280	10.94 (7.55–16.20)	0.96 ± 0.11	1.0
	AR	225	0.46 (0.29–0.71)	0.84 ± 0.10	0.04
	PR	225	5.57 (2.27–14.76)	2.02 ± 0.20	5.8
Etoxazole	S	240	4.65 (2.22–10.49)	1.07 ± 0.13	1.0
	AR	210	>1500	-	>332.6
	PR	225	>1000	-	>215.1
Milbemectin	S	225	0.51 (0.12–2.60)	1.83 ± 0.30	1.0
	AR	135	1.81 (1.11–3.27)	0.95 ± 0.20	2.3
	PR	180	0.20 (0.15–0.26)	1.33 ± 0.09	0.4
Pyridaben	S	180	0.36 (0.28–0.47)	1.31 ± 0.14	1.0
	AR	225	>1000	-	>2777.8
	PR	225	>2000	-	>5555.6
Pyflubumide	S	180	1.35 (0.94–2.08)	0.85 ± 0.11	1.0
	AR	225	0.65 (0.39–1.03)	0.76 ± 0.09	0.5
	PR	210	0.35 (0.25–0.50)	0.96 ± 0.12	0.3
Spirodiclofen	S	225	563.12 (482.69–667.24)	2.94 ± 0.33	1.0
	AR	180	1399 (1048–1960)	1.80 ± 0.21	2.5
	PR	210	1243 (937.04–1780)	0.92 ± 0.15	2.2
Spiromesifen	S	240	2.94 (0.87–5.53)	1.19 ± 0.21	1.0
	AR	210	>1500	-	>510.2
	PR	225	473.22 (308.49–786.59)	0.65 ± 0.07	161.0

^(a)^ CL, Confidence limit. ^(b)^ RR, resistance ratio = LC_50_ of resistant strain/LC_50_ of susceptible strain.

**Table 3 insects-11-00511-t003:** Susceptibility to acaricides in the S, AR and PR strains of *Tetranychus urticae* eggs.

Acaricides	Strain	N	LC_50_ (mg/L) (95% CL ^(a)^)	Slope ± SE	RR ^(b)^
Abamectin	S	850	0.88 (0.64–1.24)	1.58 ± 0.18	1.0
	AR	668	1.35 0.94–7.97)	1.71 ± 0.22	1.5
	PR	554	0.83 (0.39–1.69)	1.54 ± 0.26	0.9
Acequinocyl	S	553	1.48 (0.56–6.37)	1.06 ± 0.19	1.0
	AR	668	189.71 (117.52.-281.56)	1.99 ± 0.18	128.2
	PR	881	1.47 (0.53–3.10)	1.42 ± 0.21	1.0
Azocyclotin	S	1140	7.69 (5.63–10.78)	1.69 ± 0.19	1.0
	AR	1041	19.54 (13.43–26.48)	1.17 ± 0.22	2.5
	PR	764	87.45 (78.35–97.27)	3.53 ± 0.37	11.4
Bifenazate	S	855	4.01 (1.46–8.74)	1.16 ± 0.17	1.0
	AR	437	4.34 (1.97–7.80)	0.98 ± 0.11	1.1
	PR	648	2.89 (0.84–6.78)	0.84 ± 0.10	0.7
Cyenopyrafen	S	736	0.76 (0.40–1.34)	1.61 ± 0.26	1.0
	AR	964	0.48 (0.17–1.21)	0.85 ± 0.11	0.6
	PR	886	1.55 (0.75–3.40)	1.24 ± 0.17	2.0
Cyflumetofen	S	892	0.58 (0.29–0.99)	1.83 ± 0.31	1.0
	AR	762	4.64 (2.22–8.67)	1.21 ± 0.15	8.0
	PR	563	4.87 (2.67–9.23)	1.39 ± 0.17	8.4
Etoxazole	S	1401	0.08 (0.06–0.11)	1.35 ± 0.14	1.0
	AR	310	0.92 (0.41–1.85)	1.06 ± 0.13	11.5
	PR	284	>500	-	>6250
Milbemectin	S	525	0.09 (0.05–0.13)	1.24 ± 0.09	1.0
	AR	754	0.28 (0.17–0.54)	0.07 ± 0.09	3.1
	PR	469	0.61 (0.22–2.75)	0.92 ± 0.14	6.8
Pyridaben	S	462	0.73 (0.31–1.48)	0.91 ± 0.10	1.0
	AR	447	>500	-	>684.9
	PR	619	>2000	-	>2739.7
Pyflubumide	S	1015	0.05 (0.02–0.09)	1.28 ± 0.15	1.0
	AR	843	0.16 (0.06–0.44)	1.08 ± 0.13	3.2
	PR	785	3.05 (2.01–4.66)	0.75 ± 0.07	61.0
Spirodiclofen	S	742	16.47 (13.39–19.51)	3.46 ± 0.38	1.0
	AR	957	79.20 (68.19–92.45)	4.31 ± 0.46	4.8
	PR	667	18.89 (13.32–28.24)	0.85 ± 0.11	1.1
Spiromesifen	S	1104	0.51 (0.29–0.80)	2.01 ± 0.27	1.0
	AR	1012	0.34 (0.19–0.58)	1.37 ± 0.17	0.7
	PR	876	1.03 (0.81–1.34)	0.94 ±0.02	2.0

^(a)^ CL, Confidence limit. ^(b)^ RR, resistance ratio = LC_50_ of resistant strain/LC_50_ of susceptible strain.

**Table 4 insects-11-00511-t004:** Genotypes of point mutations in the *cytb* and PSST genes of *Tetranychus urticae.*

Strain	N	*cytb* Genotypes (%)								PSST Genotypes (%)	
		G126S		A133T		I256V		N321S		H92R	
		G	S	A	T	I	V	N	S	H	R
S	200	99	1	100	0	98	2	100	0	90	10
AR	200	96	4	100	0	19	81	0	100	12	88
PR	200	98	2	99	1	99	1	99	1	6	94

**Table 5 insects-11-00511-t005:** Detoxifying enzyme activities in S, AR, and PR strains of *Tetranychus urticae ^(a)^.*

	S	AR			PR		
		Activity ± SD	Ratio	*p* ^(b)^	Activity ± SD	Ratio	*p* ^(b)^
**Glutathione *S*-Transferase**							
CNDB	30.3 ± 4.9b	68.2 ± 11.0a	2.3	0.0083 **	35.8 ± 5.3b	1.2	0.4679
DCNB	1.0 ± 0.3bc	1.8 ± 0.1a	1.8	0.0206 *	1.6 ± 0.4ab	0.9	0.0559
NBC	1.4 ± 0.7b	3.1 ± 0.7a	2.2	0.0044 **	1.4 ± 0.6b	1.0	0.9734
**Nonspecific Esterase**							
α-NA	626.6 ± 51.5ab	554.3 ± 55.9b	0.9	0.3340	741.3 ± 56.3a	1.2	0.1482
β-NA	686.5 ± 36.0bc	583.4 ± 39.1c	0.8	0.0751	771.2 ± 46.7ab	1.1	0.1890
**P450**							
TMBZ	1623.3 ± 209.8a	1562.0 ± 226.9a	1.0	0.8155	1763.2 ± 200.7a	1.1	0.6150

^(a)^ Means within a column followed by a different letter are significantly different. ^(^^b)^
*t*-test; * *p* < 0.1; ** *p* < 0.01 [25].

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
