# Peer review of "Target-Site Mutations and Glutathione S-Transferases Are Associated with Acequinocyl and Pyridaben Resistance in the Two-Spotted Spider Mite Tetranychus urticae (Acari: Tetranychidae)"

_insects, 2020, doi:10.3390/insects11080511_

Round 1
Reviewer 1 Report
In this paper, the authors evaluate the susceptibility of laboratory-selected strains of Tetranychus urticae to several insecticides by analyzing their target-site mutations and the involvement of GSTs. By using toxicity bioassays, genetic analyses and enzymes assays, the authors found evidence of cross-resistance, showing in some cases high level of resistance. Insecticide resistance is an important concern for pest control, because it is seriously hampering chemical control strategies, therefore studies that are focused to understand the mechanisms underlying this phenomenon are welcome. The manuscript is generally sound, but some information about methods should be included. Below my specific comments.
Comments
Lines 82-90: The authors described the procedure, but they did not mention how many disks were used and how many individuals were finally used for these experiments. This information should be added in this section.
Lines 107-115: The authors should give more details about how they performed these analyses, for examples the number of individuals that were used. A more detailed description of the procedure can be important, also considering that they did some adaptations starting from the methods of Salman and Saritas.
Lines 106-129: Again, how many individuals were used for the RNA extraction? RNA was extracted from single individual or pools? Include this information in the methods.
Lines 136: Here, it is stated that the Resistance Ratio was calculated not only in adults, but also in the eggs. However, in the section describing the toxicity bioassays, the eggs were not mentioned. Did you performed toxicity bioassay using also this stage? If so, as it seems by the results, it should be stated in the methods, describing also how these experiments were done.
Table 5: delete the asterisk of beta-NA
Lines 174-179: The results of GSTs analysis by qRT-PCR showed that only delta GSTs were up-regulated, and only in the AR strain. How many genes were analysed for each class in the two strains? It is not clear. This information should beexplicated because if only one or few genes in each class were analysed, the authors cannot rule out that other genes, that they have not analyzed, may be overexpressed. This implies that other GSTs families may be involved in the two strains. The authors should better specify this point and, eventually, discuss this aspect.
Conclusions: As presented in this version, this section is maybe too short to stand alone. I suggest to the authors to include the sentence in the Discussion or, alternatively, to extend the Conclusions, highlighting more general implications of their results in the study and/or managing of insecticide resistance.
Author Response
We appreciate the consideration of the three anonymous reviewers and the final correction. All suggestions were accepted, and the appropriate revisions were made.

Reviewer 2 Report
This manuscript described the studied results of mechanisms associated with the acequinocyl- and pyredaben-resistant two-spotted mites, and the factor results in the cross resistance of acequinocyl to pyredaben. Through the bioassay, PCR/sequencing, and enzyme activity assays/real time PCR, the authors concluded that “the I256V and N321S mutations and the increased GST metabolism and GST delta expression might be related to acequinocyl resistance in T. urticae”in the Conclusions. In addition, in the Discussion mentioned that “the H92R mutation is thought to be the reason for cross-resistance to pyredaben in the AR stain of T. urticae”.
The observations are interesting, however, there are two weak points in the abstract. First, the conclusion of the I256V and N321S mutations in cytochrome b and the increased GST metabolism and GST delta expression are associated with acequinocyl resistance in T. urticae. Second, the last sentence in the abstract “These results suggest that the AR strain 26 pyridaben cross-resistance may be influenced by the H92R point mutation and GST activity” included wrong information. “and GST activity” should be removed.
Beside the abstract, the Discussion only focused on the resistance and cross resistance between AR and PR strains of T. urticae. In fact, full of information in the bioassay results can be discussion and applied to the resistance management of T. urticae. I suggest that the authors might also discuss the positive and negative cross resistance of AR and PR strains of T. urticae to acaricides with different mode of action, and how to manage the resistant problem of field T. urticae according to these results.
Author Response
All suggestions were accepted, and the appropriate revisions were made.
Thank you so much for your kind comments.

Reviewer 3 Report
Overview
The manuscript “Target-Site Mutations and Glutathione S-Transferases are Associated with Acequinocyl and Pyridaben Resistance in the Two-Spotted Spider Mite Tetranychus urticae (Acari: Tetranychidae)” by Choi et. al provides a characterization of pesticide resistance in several populations of T. urticae. The authors characterize two insecticide-selected strains by measuring cross resistance, sequencing target sites, and performing in vitro assays to measure xenobiotic metabolizing enzymes.
The paper is generally well thought out and was clearly a large amount of work. It will no doubt be of interest to the resistance community. However, there are some concerns about the clarity of the manuscript, and it may be difficult for non-experts to interpret the results. I therefore, author address several issues prior to publication.
Major Comments
Clarity about acaricides, proteins, and mutations
It was difficult to for a researcher who does not work on these particular insecticides (myself) to interpret the results because the target site of each acaricide and substitution was not always stated. This is particularly confusing in lines 45-49 with the mention of G126S+A133T and I256V+N321S in quick succession without stating which proteins these substitutions belong to. Furthermore what protein contains the H92R mutation? Lines 188-190 and 201-209 of the discussion describes the distinct modes of action of each compound clearly, and the physiology of these target proteins within the electron transport chain. It would be beneficial to have this stated explicitly in the introduction.
Furthermore, other parts of the introduction would benefit from further elaboration, so that the reader is more familiar with the different aspects of the paper. Xenobiotic detoxification and resistance monitoring seem to be major topics in the manuscript, but are each only mentioned in 2 sentences in a paragraph together with target-site resistance. While an extensive literature review is not required additional examples of known detox genes from spider mites or monitoring programs would be beneficial.
The discussion also presents several problems. In lines 190-193 the cross resistance between chlorfenpyr and bifenazate reported by Van Leeuwen et. al has no context as the reader cannot be assumed to know that the targets of both of these compounds. The same could be said for all of the data reported in Tables 2 and 3 which is not discussed at length in the paper. Does cross resistance arise from compounds with the same mode of action (implying target site resistance)? Does cross resistance arise from compounds with the different modes of action (implying other mechanisms)? From my knowledge both would be the case and it would be useful to mention this in the manuscript. Perhaps another column could be added to table 2 and 3 to describe the mode of action of each compound.
qPCR on GSTs
It is not clear in the current manuscript which gene(s) the qPCR primers listed in table 1 target. From my understanding there are several GST genes in each of the subclasses. were primers designed to be common to all of these genes or were specific genes within each subclass chosen? If the latter is true, which genes were chosen and why?
Genetic background
It should be mentioned more explicitly in the text that the S, AR, and PR were from completely distinct genetic backgrounds. While useful to compare resistant and susceptible populations the strength of the study would be improved if more comparable backgrounds were used (e.g. a subset of the AR and PR populations that did not undergo selection). While this is not always possible one sentence mentioning this is necessary to interpret the results properly.
Minor Comments
Line 17: It needs to be mentioned what these resistance ratios refer to in terms of both pesticide and genetic background. In other words, which two strains are being compared (S vrs AR; S vrs PR) with which acaricides?
Line21-22: The sentence on GST cross resistance must be modified to soften the statement. E.g. “In some cases increased GST activity has previously been linked to enhanced detoxification”
Line 25-26: It needs to be clearer which strains are being compared. The delta class GST(s) in the AR strain were significantly higher than both the PR and susceptible background (Figure 4).
Lines 35-37: The fact that a “deacetylated metabolite of acequinocyl inhibits respiration in mitochondria” mean that this compound is a prodrug? If so, it would be good to state this explicitly to avoid any confusion from readers.
Line 47: “point mutations” (add s)
Line 49: “point mutations” (add s)
Line 67-69: It is too strong a statement to say that the S strain has never been exposed to any insecticides as it may have been exposed prior to 2005.
Line88-89: What formula was used for correcting mortality data?
Line 95-99: It would improve the clarity of this section (2.4) if the authors describe what these primers are targeting. What regions of which genes are they targeted to amplify? Also, it would be good to include the sequences of previously published primers in a supplementary file for ease of access
Lines 124-125: From my understanding qRT-PCR typically does not measure ΔΔCT at the end of 40 cycles but rather at a defined threshold where the amount of product is separated from statistical noise. If the authors have deviated from the normal methodology, they should explain this decision in the methods section. The reference gene (Actin?) should also be stated explicitly in the text.
Line 164: With what frequency was the H92R mutation found in the PR strain? While Table 4 indicates this, it should also be mentioned in the text.
Line 185: “yr” should be changed to “years”
Line 200-201: Where was the I256V and N321S variants found in citation 19 found? This is important to mention in terms of determining whether these mutations have arisen independently or were introduced to the AR and PR populations via gene flow.
Figure 1: It would be helpful to add titles to each of the panels, so that the reader does not have to refer to the legend for each figure
Recommendations (optional)
The cross-resistance data that the authors generated will no doubt be of interest to readers. However, I found it difficult to easily see because it was all in a table. While table’s are necessary to communicate exact values, the inclusion of a figure would greatly facilitate the ability to pick trends out of the data. The same is true of Table 5. These would not replace the figures, but it could mean that the table’s move to supplementary data if the authors and editor wish.
Author Response

(The authors gave the same response as above.)

Round 2
Reviewer 1 Report
In the new version, the authors amended the text, improving the clarity of the paper, especially regarding the experimental procedure. I suggest to the authors to include in the maintext of the manuscript (in the paragraph 2.6) how they selected the genes that were anlysed by qRT-PCR, specifying also how many genes were analysed.
Minor comments:
Lines 94-114: change "discs" with "disk".
Line 268: “We selected only one gene in each GST class. So, additional research is needed to confirm”. Make a unique sentence.
Author Response
All suggestions were accepted, and the appropriate revisions were made.

Reviewer 3 Report
Overview:
The authors have done a good job at addressing the points outlined by the reviewers. As a result I would be happy to recommend the manuscript for publication. However, there are some remaining issues with the new edits and a few cases where the previous recommendations were not adequately incorporated.
Minor comments:
Line 18: “S” needs to be defined before being used as an abbreviation
Line 26 & Line 217: It must be said that only one gene from each GST class is used. As it is currently written it appears as though multiple genes from each class were tested. The authors have corrected this in Line 268-269, but it must be stated prior to that as well.
Line 49: In what gene/found are the (G126S+A133T) mutations found?
Line 166: The authors have not provided an explanation or amendment regarding the qPCR methodology. From my understanding qPCR does not measure ΔΔCT at the end of the reaction (40 cycles), but at a defined threshold during the reaction. The authors must explain this choice (perhaps I am wrong) or amend the text. In any case this result is peripheral to the paper, so no further experimentation is needed.
Line 230-231: The sentence in these lines appears to repeat the same idea as the sentence before it.
Line 269: “additional research is needed to confirm or reject our findings”
Author Response

(The authors gave the same response as above.)

Round 3
Reviewer 3 Report
The authors have made all required changes to the manuscript.